# Can the Nasal Cavity Help Tackle COVID-19?

**DOI:** 10.3390/pharmaceutics13101612

**Published:** 2021-10-03

**Authors:** Bissera Pilicheva, Radka Boyuklieva

**Affiliations:** 1Department of Pharmaceutical Sciences, Faculty of Pharmacy, Medical University of Plovdiv, 4002 Plovdiv, Bulgaria; radka.boyuklieva@phd.mu-plovdiv.bg; 2Research Institute at Medical University of Plovdiv, 4002 Plovdiv, Bulgaria

**Keywords:** intranasal delivery, nasal cavity, nasal vaccine, SARS-CoV-2, COVID-19

## Abstract

Despite the progress made in the fight against the COVID-19 pandemic, it still poses dramatic challenges for scientists around the world. Various approaches are applied, including repurposed medications and alternative routes for administration. Several vaccines have been approved, and many more are under clinical and preclinical investigation. This review aims to systemize the available information and to outline the key therapeutic strategies for COVID-19, based on the nasal route of administration.

## 1. Introduction

Since its outbreak in December 2019, the corona virus disease (COVID-19) has remained a challenging topic in medicine. Society has been facing numerous challenges the regarding spreading of the disease, health prevention, reduction of deaths, etc. Therapeutic programs have been changing constantly, and various approaches have been applied. Currently, COVID-19 remains a serious health risk despite the significant advances accomplished by the global pharma sector. The number of confirmed cases as of 15 August 2021 approaches 200 million, according to WHO data, and the confirmed deaths are more than four million [1]. About a third of those people who are infected do not develop noticeable symptoms [2]. Of the people who develop symptoms, most are not severely affected and develop mild-to-moderate symptoms. However, a cohort of almost 5%, most of them being older people, suffer very critical and life-threatening symptoms. Some people experience long-term effects for months after recovery, most often with serious damage to essential organs, such as heart, kidneys, lungs, brain, etc. [3].

Despite the global efforts to bring the health professionals and scientists together to accelerate the R&D process in diagnostics, immunization or therapeutics, no specific solution has been reached so far. Management of the disease involves symptomatic or supportive care, preventive measures to minimize the risk of transmissions and experimental measures. A great load of work is underway to develop virus-inhibiting drugs; several vaccines have been approved, and mass vaccination programs have been initiated worldwide. As the COVID-19 pandemic is ongoing, it is of paramount importance to develop effective approaches for limiting further spread of the disease. Despite the global consolidated efforts and accomplished results, COVID-19 remains a threat to public health. The purpose of this article is to review the key therapeutic strategies for COVID-19 based on the nasal route of administration.

## 2. The Nasal Cavity as an Entry Point for SARS-CoV-2

COVID-19 is mainly transmitted via the respiratory route after inhalation of contaminated droplets or particles [4]. Current studies reveal that, after entering the body, severe acute respiratory syndrome coronavirus-2 (SARS-CoV-2) spreads to the back of the nasal passage and to mucous membranes in the throat, attaching to the body’s cell receptors [5]. One major receptor for SARS-CoV-2 is angiotensin-converting enzyme 2 (ACE2) receptor protein, which is widely expressed in the cells of the lungs, intestine, liver, heart, vascular endothelium, testis and kidney [6,7,8].

In a recent study by Sungnak et al. [9], SARS-CoV-2 entry factors were thoroughly investigated, and the results confirmed that ACE2 protein was expressed in multiple tissues, as previously published in the literature [10,11,12].

Additionally, it was found that these factors are highly expressed in tissues not previously investigated, including nasal epithelial cells. The findings imply that the nasal cavity may have primary role in viral transmission through contaminated airborne droplets. Furthermore, SARS-CoV-2, being an envelope virus, might exploit existing secretory pathways in nasal goblet cells, because it does not require cell lysis for releasing [9].

In a study by Chen et al. [13] immunohistological analysis was performed to determine the location of ACE2 protein in human nasal and tracheal specimens. High levels of ACE2 protein in the nasal cells located in the olfactory neuroepithelium (200 to 700-fold increase compared to other nasal cells) were detected. Since this area is rich in odor-sensing neurons, the researchers suggest that infection of these cells could be the reason for olfactory dysfunction in COVID-19 patients [14].

Furthermore, a retrospective examination of nasal epithelium from individuals aged 4 to 60 years was conducted, focusing on the gene expression of ACE2 protein. Age-dependent ACE2 gene expression in nasal epithelium was found: the lowest ACE2 gene expression was estimated in younger children and increased with age, thus confirming the hypothesis that the lower risk among children is due to differential expression of ACE2 in nasal epithelium, the first point of contact for SARS-CoV-2 and the human body [15].

In view of the above discoveries, drugs or vaccines administered intranasally are likely to target specifically those cells and therefore could be highly effective in limiting the spread.

## 3. Prevention of COVID-19 via the Nasal Cavity

### 3.1. Nasal Delivery of Vaccines

Mucosal vaccination has gained much attention in the recent years as an alternative to conventional injections. Different mucosal routes have been investigated, but the nasal route has been preferred for its numerous benefits. As a mucosal route of vaccination, it induces both local immune response and systemic immunity, opposed to traditional parenteral administration, which generally results only in systemic immune responses. Moreover, nasal vaccination is believed to result into immune protection in distant mucosal organs. Nasal vaccination serves as a blockade of pathogens’ entry, inducing specific immune response in the mucosal tissue.

The inductive site for mucosal immunity in the nasal cavity is the nasal-associated lymphoid tissue (NALT) [16]. This concept is based on findings in rodents; however, a study has documented the presence of NALT in humans as a morphologically distinct structure additional to the lymphoid structures of the Waldeyer’s ring [17,18]. The NALT consists of lymphocytes, B cells, T cells and antigen-presenting cells (APCs) and is defined as organized mucosal immune system in the nasal mucosa [19]. It is covered by an epithelial layer containing memory (M) cells, which are responsible for antigen intake from mucosa [20,21]. In addition, fast vaccine absorption in the circulatory system is achieved, which increases the overall efficacy of the vaccine [22].

The nasal route of administration is considered highly effective for vaccination, due to multiple advantages. The nasal cavity is an easily accessible organ, providing needle-free vaccination. Furthermore, nasal mucosa is highly vascularized and supplied with numerous microvilli, providing a large surface area that is available for absorption and fast onset of strong immune response [23]. Given the upper considerations, better patient compliance is achieved due to convenience, cost, ease of administration and disposal, which is essential for successful vaccination.

### 3.2. Challenges and Considerations in the Development of Nasal Vaccines

Nasal vaccines must be specifically formulated and optimized to achieve an adequate immune response and prevent from local irritation or other potential side effects. Administration of a nasal vaccine onto the mucosal tissue may result into dilution of antigens by nasal secretion, seizure in the mucus gel, inactivation by nasal enzymes or blockade by epithelial barriers [24]. Furthermore, normal defense mechanisms, such as mucociliary clearance and ciliary beating, may limit the retention time of the formulation on the nasal mucosa resulting into inefficient uptake of soluble antigens by nasal epithelial cells in the nasal cavity [25,26,27]. Additionally, high-molecular-weight compounds are not easily delivered via the nasal route, and large doses of vaccine are required to achieve optimum immune response, which is challenging, since the nasal-cavity volume capacity is restricted to 25–200 μL [28]. Moreover, adjuvants may be needed to enhance the nasal vaccine immunogenicity and delivery to the mucosal tissues.

Various strategies have been outlined for the development of safe and effective vaccine formulations for nasal administration. Most of them are focusing on the use of advanced delivery systems targeting specific regions of the nasal cavity, improving mucoadhesion and residence time and ensuring enhanced stability of the formulation [29,30]. Pharmaceutical dosage form is another substantial factor to consider when developing a nasal vaccine formulation [31]. Current options for delivery of nasal formulations include solutions (drops or sprays), powders, gels and solid inserts. The selection of a pharmaceutical form depends mainly on the antigen used for the proposed indication, patient type and marketing preferences [32].

Solutions are considered the easiest way to formulate a vaccine and the most convenient dosage form for administration in the nasal cavity. Disadvantages of such systems include lack of dose precision, tendency to degrade and a short residence time. A wide variety of natural, synthetic and semi-synthetic polymers have been investigated for their capacity to enhance mucoadhesion and prolong residence time of the formulation at the application site [33,34].

Powder formulations have been widely studied for their capacity to enhance vaccine stability. Moreover, nasal powders can prolong the contact time for powder formulations on the nasal mucosa, inducing stronger local and systemic immune response. However, the production of nasal dry powders with optimum characteristics (uniform particle size and size distribution, flow property and performance) is quite complicated, and special application devices may be needed [35]. Since soluble antigens are less immunogenic than particulate formulations, encapsulation of antigens into particles is expected to facilitate the antigen transport across the nasal mucosa and thus improve its uptake. For this reason, there has been a growing interest towards designing particulate systems as carriers for vaccines [36,37,38].

Particulate delivery systems that can mimic pathogens, such as polymeric particles, immune-stimulating complexes (ISCOMs) and liposomes, are considered promising for nasal vaccine delivery [39,40,41]. Particulate antigens reach the lymphoid tissues via transcellular route and target M cells. Since they have a similar size to pathogens (below 5 µm), a natural infection is imitated. The immunogenicity of liposomes is mainly due to their capacity to accommodate multiple copies of antigens, to be preferentially taken up by macrophages [42]. ISCOMs, which consist of an adjuvant (saponin), lipids and an antigen, held together by hydrophobic interaction [43], have been widely investigated for their immunostimulatory potential conferring induction of specific local and systemic immune responses following intranasal (IN) administration [44,45,46]. Immunogenicity of antigens may be significantly improved by using adjuvants, which can also act as delivery systems. These may also reduce the amount of antigen required to induce an immune response. Particulate delivery systems tend to combine both the benefits of optimized delivery across mucosal tissue and inherent adjuvanting effects, which has been confirmed in multiple studies [33,47].

Currently, several intranasal vaccines for humans are licensed, including the influenza vaccines FluMist/Fluenz™ (MedImmune, Gaithersburg, MD, USA) [48] and the Nasovac™ influenza nasal spray (Serum Institute of India, Pune, India) [49,50]. Today, with increased need to immunize large populations, potentially in swift response to pandemics, such as COVID-19, there is a clear need to have strategies in place.

### 3.3. Intranasal Vaccines against SARS-CoV-2 in Clinical Trials

It is of crucial importance to provide access to safe and effective vaccines to end the COVID-19 pandemic. As per August 2021, according to WHO COVID-19 Landscape of novel coronavirus-candidate-vaccine development worldwide data [51], 294 vaccine candidates are under development, 110 of which are in the clinical phase. Of all candidates in the clinical phase, eight (7%) are designed for intranasal administration, one is intramuscular or intranasal vaccine and one is with intramuscular first dose, followed by intranasal booster. Of 184 vaccines against SARS-CoV2 that are in preclinical development, eight are designed for intranasal administration (Table 1).

#### 3.3.1. Viral Vector Vaccines

The success of virus-based vector vaccines is due to their ability to infect cells, which makes them capable of eliciting a strong immune response. They are characterized by high-efficiency gene transduction and specific delivery of genes to target cells. However, they are constrained by disadvantages, such as low titer production, risk of inducing anti-vector immunity and generation of replication-competent virus, which may cause tumorigenesis. One of the main obstacles for production of viral vector vaccines is scalability. Generally, vectors are grown in cells that are attached to a substrate, rather than in free-floating cells, which is difficult to be achieved on a large scale. Suspension cell lines are now being developed, which would enable viral vectors to be grown in large bioreactors. Assembly of the vector vaccine is also an elaborate process, involving multiple components and steps, each of which increases the risk of contamination. Extensive testing is therefore required after each step, generally resulting in increased costs [52,53].

##### ChAdOx1-S

ChAdOx1-S (other names AZD1222, Covishield, Vaxzeriva), developed by Astra Zeneca and Oxford University, is a replication-deficient simian adenovirus vector vaccine, given by intramuscular injection. To assess the efficacy of an IN vaccination with ChAdOx1-S three groups of 10 Syrian hamsters were vaccinated with a single dose: group I received ChAdOx1-S via IN route, group II received the same dose of vaccine via IM route and group III received control vaccine ChAdOx1 GFP via IM route. Four weeks after vaccination, the animals were given SARS-CoV-2/human virus intranasally. For the testing of transmission, vaccinated animals were housed with non-vaccinated donor animals for 4 h. Regardless of the route of vaccination, high IgG titers with no significant difference were obtained. Neutralizing antibodies were significantly higher in IN vaccinated animals. Moreover, there was a significant difference in the viral RNA and viral load in the oropharyngeal swabs for IN vaccinated compared to control animals, while no difference in the amount of viral RNA and infectious virus was noted for IM vaccinated animals as compared to the control. No viral RNA or infectious virus was detected in lung tissue from IN vaccinated animals [54,55]. IN vaccination in rhesus macaques by means of a mucosal atomization device was performed. The formulation was delivered in the nasal cavity in the form of sprayed aerosol. As with the previous experiment that used a small-animal model, the animals received SARS-Cov-2/human virus particles both intratracheally and nasally. Higher fractions of IgA antibodies were detected in the nasal swabs compared to bronchoalveolar lavage (BAL) fluid and serum samples. Higher IgG titers were found after the booster vaccination (28 days post-infection). More importantly, SARS-CoV-2 specific IgG and IgA were found in BAL and nasosorption, demonstrating that IN vaccination induced systemic immunity comparable to that after IM vaccination, but also elicited SARS-CoV-2-specific mucosal immunity as demonstrated by IgA detection in nasosorption and BAL samples [56]. On 1 April 2021, the University of Oxford commenced a Phase I clinical trial to investigate the safety and tolerability of intranasal administration of ChAdOx1-S. The purpose of this study is to test a new route of administration for the Oxford/AstraZeneca COVID vaccine in healthy volunteers aged 18–55 years. The trial is expected to provide valuable information on the safety of the vaccine and extent of the immune response when administered intranasally [57].

##### DelNS1-2019-nCoV-RBD-OPT1

The University of Hong Kong, Xiamen University and Beijing Wantai Biological Pharmacy developed the DelNS1-2019-nCoV-RBD-OPT1-viral vector replicating vaccine. It is based on genetically engineered live attenuated influenza virus without adjuvants, to express the receptor binding domain (RBD) of SARS-CoV-2’s spike protein for triggering immune responses against SARS-CoV-2. On 1 September 2020, a Phase I clinical trial was initiated in China to evaluate the safety of DelNS1-2019-nCoV-RBD-OPT1 vaccine [58]. The study enrolled 115 participants between 18 and 55 years. The same vaccine candidate has also been registered for a Phase II trial whose primary purpose is to evaluate its immunogenicity according to different immunization procedures. Serum total antibodies, IgG antibodies and neutralizing antibodies of SARS-CoV-2 one month after the last dose of vaccination, as well as the specific cellular immune response level of SARS-CoV-2 spike protein in whole blood 14 days after the last dose of vaccination will be tested as primary indicators. Local and systemic adverse reactions within 14 and 42 days after each dose, and serious adverse events during the entire study period (12 months after the full vaccination) will be monitored [59].

##### AdCOVID^TM^

Altimmune, Inc., developed a single-dose COVID-19 vaccine, AdCOVID^TM^, based on a replication-deficient adenovirus type 5 (Ad5) vectored vaccine encoding for the receptor binding domain (RBD) of the SARS-CoV-2 spike (S) protein. The formulation was designed for intranasal administration and was attributed with key potential benefits, including simple nasal delivery, scalable manufacturing and high room temperature stability for several months. In a preclinical study on mice, testing the immunogenicity of AdCOVID^TM^ after intranasal administration of the vaccine, quite promising results were obtained. Strong IgG serum neutralizing activity was demonstrated, exceeding several times FDA recommended titers. Mucosal immunity was also potent, with a 29-fold increase in mucosal IgA measured in the BAL fluid [60]. In February 2021, Altimmune commenced its Phase 1 clinical trial of AdCOVID^TM^ to investigate the safety and immunogenicity of the intranasally administered vaccine candidate in approximately 80 healthy volunteers aged 18–55. Participants were given one or two doses of AdCOVID^TM^ as a nasal spray, at three dose levels. In addition to the primary study endpoint of safety and tolerability, the immunogenicity evaluation included serum binding and neutralizing antibody titers and mucosal IgA antibody from nasopharyngeal swabs post-vaccination. The immunogenicity data demonstrated weaker-than-expected immune responses for each of the immune parameters tested. Although AdCOVID^TM^ was able to stimulate antibodies that bound the SARS-CoV-2 spike protein and were able to neutralize the virus, the magnitude of the immune activation and the share of respondents were not satisfactory. Based on these data, Altimmune has announced discontinuing further development of AdCOVID^TM^.

##### BBV154

It is a simian adenoviral (ChAd36) vector-based (expressing a stabilized spike protein) intranasal vaccine, which stimulates a broad immune response—neutralizing IgG, mucosal IgA and T-cell responses. The vaccine is developed by Precision Virologics, a startup incubated at the Washington University School of Medicine (St Louis, MI, USA) in collaboration with Bharat Biotech International Limited (Hyderabad, India), which has already marketed IM COVID-19 vaccine named Covaxin^TM^ (BBV152). In a study using a murine model for SARS-CoV-2 infection, intranasal administration of the vaccine was followed by a strong mucosal immune response preventing respiratory tract infection and inflammation [61,62]. To compare the efficacy of nasally delivered ChAd-SARS-CoV-2 vectors with IM administration, the vaccine was tested on a Syrian hamster model. Six-fold higher titers of neutralizing antibody were detected after IN vaccination, reinforced by diminished viral RNA in lungs and nasal tissues [63]. On March 1, 2021, Bharat Biotech International Limited initiated a Phase I randomized, double-blinded, multicenter clinical trial in India. The purpose of the study is to evaluate the immunogenicity and reactogenicity of BBV154 COVID-19 vaccine in healthy adults and establish its safety profile in one- and two-dose regimens. The trial is expected to be completed in November 2021 [64].

##### PIV5

Parainfluenza virus type 5 (PIV5) vector vaccine that encodes the SARS-CoV-2 spike protein (also known as CVXGA1-001) was developed by CyanVac LLC. PIV5 is a negative-stranded RNA virus, which has been evaluated as a vaccine vector for influenza, respiratory syncytial virus (RSV), rabies and a variety of other pathogens. In animal models, PIV5 is safe and is not associated with any disease, except for kennel cough in dogs. Intranasally administered kennel cough vaccines containing live PIV5 have been used for more than four decades with an excellent safety record [65,66]. In a Phase I clinical study that has started in the USA in July 2021 (CVXGA1), intranasal COVID-19 vaccine will be administered by dose escalation (low dose to high dose) and age escalation in 80 healthy adults aged 18–75 years, with an approximately 12-month follow-up. The goal of the trial is to evaluate its safety and immunogenicity.

#### 3.3.2. Live Attenuated Virus Vaccines (LAV)

These are whole-virus vaccines that use a weakened (attenuated) form of the pathogen, which causes a disease to induce immunity. Generally, attenuation is accomplished by growing the virus at adverse conditions such as unfavorable temperature or by modification of the virus genome (e.g., codon de-optimization, removal of certain genes, etc.). Despite being time-consuming and technically challenging, these approaches have proven to be cost-effective for large-scale manufacturing and facilitated regulatory approval due to estimated high efficacy and potency in multiple in vitro and in vivo experiments. Usually, long-lasting immunity is provided after a single dose [52].

##### COVI-VAC

This live attenuated intranasal vaccine was developed by Codagenix, Inc., to prevent COVID-19, using live virus platform, and applying re-coding of viral genes for the spike protein. Synthetic attenuated virus engineering (SAVE) technology was used to recover and amplify coronavirus strain WT 2019-nCoV/USA-WA1/2020 (WA1) and to designed two de-optimized LAV candidates against the strain.

The attenuation, safety and efficacy of the vaccine were evaluated on a golden Syrian hamster model. Initially, the animals received doses of 0.05 mL COVI-VAC intranasally; then, on day 16, COVI-VAC-inoculated animals were challenged IN with WT WA1. The authors of the study reported that COVI-VAC was highly attenuated and safe in these animals. Additionally, the virus was not detected in brains of COVI-VAC-inoculated hamsters and COVI-VAC inoculation did not induce weight loss or significant lung pathology in infected hamsters. Furthermore, immunosorbent assay (ELISA) was carried out to measure IgG titers against SARS-CoV-2 spike S1 in hamsters inoculated with WT WA1 or COVI-VAC. A strong anti-spike S1 antibody response was induced, and high levels of neutralizing antibodies were determined in COVI-VAC-inoculated hamsters [67]. In December 2020 a Phase I clinical trial of COVI-VAC was commenced by Codagenix and the Serum Institute of India. The purpose of the study is to assess the safety and immune response of COVI-VAC in healthy adults aged 18 to 30 years in the UK. The participants enrolled in the study are assigned randomly to receive either two doses of COVI-VAC 28 days apart, two doses of placebo (saline), or one dose of COVI-VAC and one dose of placebo. COVI-VAC or placebo is delivered to the nasal cavity in the form of nasal drops. To assess the safety of the vaccine, participants are required to keep a daily record of symptoms for 14 days after each dose. Adverse events and medication use will be recorded. Blood samples and intranasal samples will be collected to assess the immune response from the vaccine [68].

##### MV-014-212

Meissa Vaccines, Inc., developed MV-014-212, a single adjuvant-free dose, live attenuated, recombinant human respiratory syncytial virus (RSV) expressing a chimeric SARS-CoV-2 spike as the only viral envelope protein. MV-014-212 was attenuated and immunogenic in African green monkeys (AGMs). According to a preclinical study performed by Tioni et al. [69], one mucosal administration of MV-014-212 in AGMs protected against SARS-CoV-2 challenge, reducing the peak shedding of SARS-CoV-2 in the nose by more than 200-fold. MV-014-212 elicited mucosal immunity in the nose and neutralizing antibodies in serum that exhibited cross neutralization against two virus variants of concern. Studies in mice indicated that MV-014-212 vaccination generated a Th1-biased cellular immune response. A Phase I clinical trial in the USA is ongoing to assess the safety and the immunogenicity of the vaccine when administered to healthy adults aged 18 to 69 years who are seronegative to SARS-CoV-2 and have not received another vaccine against COVID-19. MV-014-212 is administered as nasal drops or a nasal spray [70].

#### 3.3.3. Inactivated Vaccines

In this type of vaccines, the genetic material of viruses has been destroyed by heat, chemicals or radiation, so they can neither infect cells nor replicate, but can still trigger immune responses. They are considered safer and more stable than live attenuated vaccines, and they can be administered to people with compromised immune system. As the pathogens are inactivated, these vaccines generally stimulate a much weaker immune response than live attenuated vaccines and usually require several doses for effective immunity to be established. The induced immune response is typically humoral. Antibody titers against the targeted antigen will diminish with time, leading to the need for a booster dose [71].

##### Live Recombinant Newcastle Disease Virus (rNDV) Vector Vaccine

Live rNDV vector vaccine (also known as Patria, NDV-HXP S) is an inactivated vaccine developed by Laboratorio Avi-Mex. The vaccine is intended for intranasal or intramuscular administration. Newcastle disease virus (NDV) is an enveloped, negative-sense, single-stranded RNA virus in the family Paramyxoviridae surpassing other viral vectors in terms of effectiveness and safety. NDV can induce large amounts of IFN-1 and impede human cells response thus assuring strong immunomodulation. NDV is considered genetically stable, since it does not undergo the genetic recombination or genetic reassortment observed for certain RNA viruses [72]. The effectiveness of NDV-vectored vaccines has already been evaluated against SARS-CoV in monkeys. African green monkeys were immunized through the respiratory tract with two doses of NDV expressing the S-glycoproteins of SARS-CoV [73]. The animals developed a high titer of SARS-CoV-specific neutralizing antibodies. Upon challenging with a high dose of SARS-CoV, a significant reduction of the viral load in the lung samples was detected in immunized animals, as compared with untreated controls. These findings served as a basis for hypothesizing that eventual IN administration of the vaccine could restrict viral replication in the nasal cavity. In a preclinical study, mice and hamsters vaccinated with NDV-HXP-S demonstrated rapid neutralization of SARS-CoV-2 and variants, suggesting strong immune response [74]. A Phase I clinical study started on 20 May 2021, to evaluate the safety and immunogenicity of three concentrations of a rNDV vaccine against SARS-CoV-2 administered by the IN and IM route to healthy volunteers. It is an open-label, non-randomized, dose-escalation study, using three doses and two schemes of administration of the vaccine and involving 90 healthy volunteers at a single research site in Mexico City [75].

#### 3.3.4. Protein Subunit Vaccines

Subunit vaccines (also known as acellular vaccines) have a very strong safety profile, because, as opposed to live vaccines that contain viral particles, they comprise purified antigenic fragments (e.g., the S protein of SARS-CoV-2) capable of stimulating immune cells, without causing the disease. Despite the good safety profile, subunit vaccines have some drawbacks. Immunogenicity is lower compared with live vaccines, which necessitates administration of a booster dose. Furthermore, production is more complicated and expensive because of strict hygiene measures needed to avoid contamination with other organisms.

Currently, there are several vaccines based on this platform undergoing preclinical studies, including SARS-CoV-2 vaccine candidates.

##### CIGB-669

CIGB 669 (also known as Mambisa) is a subunit vaccine candidate developed by the Center for Genetic Engineering and Biotechnology (CIGB), Cuba. It is based on the receptor-binding domain (RBD) of the spike protein of the SARS-CoV-2 and uses the AgnHB protein as an antigen, with the ability to stimulate the immune response at the mucosal level, the first barrier against a pathogen. CIGB 669 was designed by protein engineering, using computational methods which aimed at accomplishing maximum similarity to the SARS-CoV-2 virus. Despite the lack of data from preclinical studies on CIGB 669, a randomized Phase I/II clinical trial was initiated in December 2020 to evaluate the safety and immunogenicity in adults of two vaccine candidates, based on recombinant RBD subunits for the prevention of COVID-19 in regimens that use the nasal route of administration. The trial enrolled 88 healthy volunteers aged 19–54 in Cuba. The findings of the trial are yet to be announced. [76].

##### Razi Cov Pars

Razi Cov Pars is a COVID-19 vaccine candidate developed by Razi Vaccine and Serum Research Institute, Iran. It is a recombinant spike protein vaccine, which is developed in three doses: the first two doses are injectable, and the third dose is intranasal. Preclinical trials have not been published yet. A Phase I randomized, double blind, placebo-controlled trial was initiated on 29 January 2021, enrolling 133 healthy volunteers aged 18–55 in Iran. Participants are allocated to four study groups at random. The four groups consist of three vaccine arms, receiving 5, 10 and 20 µg/200 µL IM doses on day 0 and 21, followed by 10 µg/200 µL IN spray on day 51. The fourth group receive adjuvant only on day 0 and 21 (IM) and 51 (IN spray) [77]. The injected-inhaled recombinant coronavirus protein vaccine has proved to be safe in the first phase of human clinical test. Mild complications were observed only in some vaccine recipients, and they include headache, mild fever and injection site pain, which is normal and common in every vaccine. In April 2021 Razi Cov Pars entered a Phase II clinical trial to assess the safety and immunogenicity of the vaccine candidate. The study is randomized, double blind, placebo controlled and involved 500 volunteers (divided into two study groups) aged 18–70 in Iran. The two study groups consist of one vaccine group receiving a selected vaccine dose from Phase I, and a placebo group receiving adjuvant only. They receive an IM injection on day 0 and 21, followed by IN spray on day 51. The immunogenicity and efficiency demonstrated in the first phase were verified in the Phase II trial [78].

### 3.4. Intranasal Vaccine Candidates in Preclinical Development

Given the progress made and the persisting search of an effective tool to manage COVID-19 pandemic, it is understandable why the concept of intranasal vaccination is attracting increasing attention.

Utrecht University researchers (the Netherlands) completed a project COVAC-ND dedicated to the development of an intranasal vaccine against SARS-CoV-2 in collaboration with Wageningen Bioveterinary Research (WBVR), and the Institute for Translational Vaccinology (Intravacc). The vaccine is produced by using reverse genetics technology with Newcastle Disease Virus as the vector for expressing the spike protein of SARS-CoV-2 to induce both mucosal and systemic immunity. The efficacy and safety of the vaccine have been tested in a preclinical animal model. The results from preclinical studies are to be announced [79]. Intravacc in collaboration with Leiden Academic Center for Drug Research (the Netherlands) have also developed a nasal spray vaccine based on a proprietary outer membrane vesicle (OMV) click technology. OMVs are spherical particles (ca. 20–200 nm) that are naturally released by Gram-negative bacteria and can harbor many bacterial antigens responsible for infection and bacterial survival in the host. In the click platform, the OMVs are decorated with immunogenic peptides and/or proteins that induce a more effective immune response against newly introduced antigens [80]. In this vaccine candidate, OMVs were coupled with the recombinant spike protein of SARS-CoV-2. Promising results in preclinical studies on mice and hamsters were announced demonstrating significant mucosal and systemic immunity. Furthermore, Intravacc’s candidate has proven to be stable at 4 °C for many years, and the production is inexpensive [81].

NanoVax^TM^ innovative platform was introduced by BlueWillow Biologics (Ann Arbor, MI, USA) for the development of an intranasal vaccine candidate, coded S-2P-NE-01, in collaboration with Medigen Vaccine Biologics Corporation (Taipei City, Taiwan). NanoVax^TM^ technology platform is based on a novel oil-in-water nanoemulsion (400–500 nm size) adjuvant with established safety and immunogenicity in human clinical trials and efficacy in primary and relevant animal models [82,83,84]. Preclinical evidence was reported of the safety and potency of S-2P-NE-01 vaccine candidate supported by strong neutralizing antibody response and high IgA titers in serum and BAL samples in mice compared to intramuscular injection [85].

A Finnish company, Rokote Laboratories Finland, Ltd. (Helsinki, Finland), is working on a nasal spray vaccine targeting COVID-19. The vaccine is based on gene-transfer technology, using an adenovirus vector to deliver a cloned DNA strand. There are no other parts of the virus in the vaccine. IN delivery of the DNA causes nasopharyngeal cells to produce SARS-CoV-2 viral protein which induces an immune response. The vaccine candidate performed well in preclinical studies and is expected to enter Phase I clinical trial in Finland within a few months [86].

In a study performed by An et al. [87], the feasibility STINGa (cyclic guanosine monophosphate–adenosine monophosphate) encapsulated in liposomes as potent adjuvant for intranasal vaccination was demonstrated. According to the authors, this was the first COVID-19 vaccine non-viral candidate that can induce systemic and mucosal immunity. It was found that IN vaccination stimulated IgA responses in the lung and directly in the nasal compartment, and IgA-secreting B cells in the spleen were detected. Moreover, S-specific T-cell responses were induced in the spleen and in the lung.

In another study, a lentiviral vector vaccine candidate proved to be effective against SARS-CoV-2 in animal models. The vaccine was developed in Institut Pasteur-TheraVectys Joint Lab and is designed for IN administration. The lentiviral vaccination vector, which encodes a full-length, membrane-anchored form of SARS-CoV-2 Spike glycoprotein, induces neutralizing antibodies and T-cell responses and triggers a localized immune response in the upper respiratory, providing a high level of disease protection in mouse and hamster COVID-19 models [88].

Innovative vaccine candidate against SARS-CoV-2 has been suggested by eTheRNA Immunotherapies based on their proprietary mRNA TriMix technology. The platform represents an mRNA-based vaccine adjuvant that stimulates dendritic cells into activating a strong CD4 and CD8 T-cell response. The proposed vaccine is intended for IN administration, primarily for high-risk populations, such as healthcare workers. It is also designed to be protective against future variations of the virus by targeting conserved epitopes from the whole viral genome. A proprietary delivery platform for mRNA, using a nasal atomizer, is used. One of the most promising formulation candidates is being repurposed for clinical use in collaboration with REPROCELL. Preclinical testing of the vaccine candidate was announced in March 2021 [89].

## 4. Intranasal Delivery of Therapeutics against COVID-19

### 4.1. Nasal Delivery of Antibodies

In a recent study, Ku et al. [90] reported that a COVID-19 antibody treatment they engineered has proved to be very effective at neutralizing more than 20 variants of SARS-coV-2 in mice. The new antibody-based therapy for COVID-19 is based on the development of an immunoglobulin M (IgM) neutralizing antibody (IgM-14) as an approach to overcome the resistance encountered by immunoglobulin G (IgG)-based therapeutics. For the development of six human IgM-neutralizing monoclonal antibodies, the researchers performed antibody engineering based on CR3022 monoclonal antibody and IgG1 monoclonal antibodies (CoV2-06, CoV2-09, CoV2-12, CoV2-14 and CoV2-16). Detailed studies were further performed on IgM CoV2-14 (IgM-14), and enhanced binding, neutralization and ACE2-blocking by IgM-14 over IgG-14 was documented. The feasibility of IgM-14 for nasal delivery was investigated by tracking antibody bio-distribution in vivo. It was shown that IgM-14 was deposited predominantly in the nasal cavity for more than 96 h following single intranasal administration. Moreover, evidence for IgM-14 at the application site was found at 168 h, while the antibody distribution in circulation or other organs was minimum, indicating that nasally delivered IgM-14 mainly targets the respiratory tract with a long residence time, significantly reducing the viral load. Based on these findings, the researchers conclude that IgM-14 administered nasally can serve as a therapeutic platform for COVID-19, as well as for other respiratory viral diseases.

IGM Biosciences, a biotechnology company based in California and focused on creating and developing engineered IgM antibodies, in collaboration with the authors of the study has announced their intentions to initiate a clinical trial of the treatment under the code IGM-6268 in the third quarter of 2021.

Another biopharmaceutical company, the American Eureka Therapeutics, specialized in developing novel T-cell therapies that harness the evolutionary power of the immune system, formulated a nasal spray (InvisiMask™) intended to neutralize SARS-CoV-2 from airborne droplets and particles in the nasal cavity. The formulation consists of a designed proprietary monoclonal antibody (mAb)-EU126-M2 clone in human IgG1 format. The mAb has been engineered with a proprietary adhesion technology extending the retention time of the formulation on mucosal surfaces. Upon administration by InvisiMask™ nasal spray, antibodies adhere to the upper respiratory tract forming an invisible protective mask. Then, the antibodies neutralize the SARS-CoV-2 virus via binding to the S1 Spike (S) protein and preventing from binding to ACE2 receptor on the respiratory cells. This therapy is aimed to be used for prophylactics against contracting SARS-CoV-2 infection since it blocks SARS-CoV-2 from entering cells and triggering an infection. The results of the study are promising demonstrating protection for up to 10 h at the lowest dose of 25 μg against SARS-CoV-2 pseudoviral infection in mice exposed to the highest viral load tested (10^7^ virus particles administered intranasally). Moreover, neither the nasal cavity nor lung areas showed signs of infection in EU126-M2 antibody-treated mice 7 days after virus dosing. Moreover, EU126-M2 proved to be stable in a nasal spray formulation for up to 2 weeks, at 37 °C. By August 2021, the company is preparing an Investigational New Drug application with the FDA for a clinical trial of the InvisiMask™ [91,92].

A biotechnology company based in the UK, Tiziana Life Sciences plc, has recently completed a Phase I clinical trial of its nasal anti-CD3 human monoclonal antibody, Foralumab, in mild-to-moderate COVID-19 patients in Brazil. Due to its ability to induce systemic immunity via the respiratory or intestinal epithelium, Foralumab is the first mAb that can be administered nasally or orally. The researchers evaluated Foralumab alone, and in combination with orally administered Dexamethasone. Foralumab was administered nasally once daily for ten days. The study was conducted in collaboration with scientists from the Harvard Medical School (Boston, USA) and INTRIALS (São Paulo, Brazil) and demonstrated the safety of the nasally administered formulation, the rapid nasal delivery of Foralumab and reduced pulmonary and systemic inflammation [93]. In a press release from 23 June 2021, the company announced a collaboration agreement with FHI Clinical to conduct a Phase II clinical trial for treating moderate-to-severe hospitalized COVID-19 patients in Brazil with intranasal Foralumab. The trial is expected to serve as a proof of concept for nasal delivery of Foralumab and its safety, tolerability, and efficacy as a potent, systemic anti-inflammatory treatment for severe COVID-19. In this study, Foralumab will be administered intranasally in a dose of 100 µg, using a metered-dose atomization device.

### 4.2. Nasal Delivery of Nitric Oxide (NO)

A treatment developed by the company SaNOtize Research and Development Corp. (Vancouver, Canada) proved to be highly effective and safe in a recently completed clinical trial in the UK [94]. The therapy is also being studied in Phase II clinical trials throughout Canada and in other countries. In the first 24 h, the treatment reduced the average viral load by approximately 95%, followed by more than 99% for the next 48 h. No adverse events were recorded in the trials involving more than 7000 patients testing the self-administered treatment.

The SaNOtize treatment is based on nitric oxide, a cell-signaling nanomolecule that is produced naturally by the body, as a first-in-class drug that has proven to be a potential topical antimicrobial agent to treat a wide variety of bacterial, viral and fungal diseases, including COVID-19 [95,96,97,98,99]. The liquid formulation developed by SaNOtize represents US and EU patented Nitric Oxide Releasing Solution (NORS^TM^), a self-administered platform designed to deliver NO, that can also be adapted to create NO releasing gels or creams, while controlling the dose of NO gas that is released. NO destroys the coronavirus and impedes viral replication within the cells in the upper airways preventing it from spreading to the lungs. Additionally, NO has been shown to block the ACE2 receptor essential for the virus to infect human cells. The NO produced from NORS^TM^ is totally identical to NO naturally produced by the human body. NORS^TM^ is formulated from accessible USP grade ingredients, which allows rapid and cost-effective scaling-up. Moreover, stability testing has been performed, and a one-year shelf-life in the appropriate container has been determined.

To date, NO nasal spray is one of the few novel therapeutic treatments, outside of expensive monoclonal antibodies, that is proven to reduce SARS-CoV-2 viral load in humans. Approval for Phase III trials was granted for both prevention and early treatment separately in Canada and in several other countries, including India, Brazil, and Mexico. A recent agreement with the Indian Glenmark Pharmaceuticals for long-term strategic partnership to manufacture, market and distribute NO nasal spray for COVID-19 treatment in India and other Asian markets was announced. The novel treatment has already been authorized as a medical device in Europe and is permitted for launching in the EU. NO nasal spray has also been approved and is commercially available in Israel and Bahrain, under the name Enovid™ [100,101].

## 5. Other Potential Therapies of COVID-19 for Nasal Administration

### 5.1. Xlear Nasal Spray^TM^

Xlear nasal spray^TM^, a xylitol-based formulation by Xlear, Inc. (American Fork, USA), has been suggested to play a potential role in improving the outcome in mild-to-moderate COVID-19 patients. The conclusion is based on an in vitro study carried out by scientists from Northwestern University and Utah State University who found out the components within the Xlear nasal spray^TM^, particularly grapefruit seed extract (GSE) and xylitol, were successful in statistically significant reduction of the viral load of SARS-CoV-2 [102]. Xylitol has been reported to reduce the severity of viral infections in multiple studies [103,104,105]. Combination therapy with GSE and xylitol may prevent the spread of viral respiratory infections, not just for SAR-CoV-2, but also for future H1N1 or other viral epidemics. GSE significantly reduces the viral load, while xylitol prevents the virus attachment to the core protein on the cell wall, as evidenced by electron microscopy [106].

A newly published case-report series [107] presented three patients with minimal-to-moderate risk for morbidity and mortality from COVID-19. The study demonstrated improvement in the symptoms and reduction in the clinical outcome after the use of xylitol and GSE in the form of Xlear nasal spray^TM^, as an adjunct to their ongoing treatment. None of the patients progressed to severe disease, and the number of days to testing positive to negative via COVID-19 RT-PCR nasal swab test was reduced. The findings of the study provided a rationale for initiating a larger randomized placebo-controlled clinical trial evaluating the use of xylitol plus GSE in the form of an intranasal spray in COVID-19 patients. A randomized placebo control trial to evaluate the efficacy of Xlear vs. placebo for acute COVID-19 infection was commenced in April 2021 and is expected to end by 30 August 2021, involving 200 participants. This study aims to assess the efficacy of Xlear nasal spray^TM^ as an adjunct therapy against COVID-19. This encompasses reduction in the number of days to negativization via nasal swab PCR from the average 14 days and early improvement of symptoms.

Xlear nasal spray^TM^ is currently marketed as a nasal irrigant for cleansing and moisturizing the nasal cavity. The spray uses compounds that are already approved by regulatory authorities in the UK, Europe and the USA, including polysaccharides such as carrageenan and gellan gum.

In a study by Westover et al. [108], the virucidal potential of chlorpheniramine maleate (CPM) in a nasal spray composition currently in development as an anti-allergy medication was tested. CPM is a first-generation antihistamine that has been marketed worldwide for many years. It is widely used for its antihistamine efficacy; however, recent publications suggest that CPM has strong antiviral and anti-inflammatory activity. In the study, in vitro test of the virucidal effect of a nasal spray containing CPM as an active ingredient in addition to other excipients was performed. The chlorpheniramine spray was developed and formulated by Ferrer Medical Innovations and Xlear, Inc., and contained xylitol, glycerin and sodium bicarbonate. Considering the results of this in vitro study, together with the published data supporting CPM antiviral and anti-inflammatory activity, it could be suggested that intranasally administered CMP has a great potential in the early treatment and prevention of viral infections, especially influenza A/B and COVID-19.

### 5.2. Nasal Delivery of Fusion Inhibitory Lipopeptide

Infection by SARS-CoV-2 is initiated by membrane fusion between the viral and the host-cell membrane, a process occurring either on the cell surface, or by the endosomal membrane. This phenomenon is mediated by the viral transmembrane spike glycoprotein (S). When the virus is attached or taken up by the host cell, conformational rearrangements in S are triggered, which result into membrane fusion and viral entry. Certain peptides (Heptad Repeat domain at the C terminus of the S protein, HRC peptides) are capable of inhibiting fusion, thus preventing infection [109,110].

In a recent research, de Vries et al. [111] reported the development of lipopeptide fusion inhibitors that block this critical first step of infection. These peptides have proven to inhibit various viruses, such as human parainfluenza virus type 3, measles virus, influenza virus, etc. [112,113]. Based on in vitro efficacy and in vivo bio-distribution studies, a dimeric form was selected for further evaluation in an animal model. The animal study revealed high concentrations of nasally administered lipid-conjugated peptides both in the upper and lower respiratory tract. Moreover, the lipid could be designed so that the transition from the lung to circulation is modulated. Daily administration of the formulation in the nasal cavity prevented completely SARS-CoV-2 transmission during 24-h cohousing with infected animals, whereas untreated animals were 100% infected [111]. The proposed SARS-CoV-2–specific lipopeptide could be used for effective pre-exposure and early post-exposure intranasal prophylaxis for SARS-CoV-2 transmission in humans. It is highly stable, does not require refrigeration, can be easily administered in the nasal cavity and, thus, is suited for translation into a safe and effective nasal formulation to support the battle against the ongoing COVID-19 pandemic.

### 5.3. Nasal Administration of Polymer-Based Formulations

A nasal formulation that can provide effective protection against the COVID-19 virus was developed by researchers at the University of Birmingham [114]. The nasal spray is composed of two polysaccharide polymers. The first, carrageenan, is commonly used in foods as a thickening agent, but recent data have proved its antiviral capacity [115,116] and the other was gellan gum, generally used for its mucoadhesive and viscosity enhancing property.

Three mechanisms are involved in the inhibition of the viral infection: steric hindrance at the cell interface, adsorption of the polymer to the virus and/or physical entrapment of the virus in the thick polymer layer. The polymer adsorption is facilitated through electrostatic interactions between the cell and virus membrane [117]. The polymer thus serves as a mechanical barrier that expands the hydrodynamic volume around the virus and prevents taking up by the cell. The combination approach, coupled with the highly potent antiviral capacity of the carrageenan towards SARS-CoV-2, provides a powerful spray device with the capacity to prevent both contraction and transmission.

A study performed by Morokutti-Kurz et al. [118] demonstrated dose dependent inhibition of the cell entry of the SARS-CoV-2 spike pseudotyped lentivirus by iota-carrageenan. Furthermore, iota-carrageenan showed an equivalent efficacy against SARS-CoV-2 as a neutralizing antiserum and soluble ACE2 receptors, both established parameters for clinical performance. The obtained data suggested that the use of iota-carrageenan either for prophylaxis or for therapeutic purposes may be equally effective in humans suffering from COVID-19. Clinical data and post-market-surveillance data showed that iota-carrageenan is well-tolerated, and the number of reported adverse events is very low. The authors conclude that iota-carrageenan may serve as a first broadly active treatment to close the gap between virus identification and successful developments of vaccines or specific antiviral medication in the case of the emergence of a new respiratory virus and pandemic threat.

The USA-based Amcyte Pharma announced results of a multicenter, placebo-controlled trial in which its iota carrageenan nasal spray Nasitrol reduced COVID-19 infection among unvaccinated staff who care for COVID patients, compared with the placebo.

Meanwhile, a research group at Swansea University investigated Carragelose^TM^ capacity for prevention of COVID-19 or at least for reducing severity of symptoms. Carragelose^TM^, a patented version of iota-carrageenan, has been widely tested and proved to shorten the duration and severity of cold and flu-like symptoms. A new clinical study performed in collaboration with Marinomed Biotech AG (Austria) revealed that Carragelose^TM^ could also reduce the risk of SARS-CoV-2 infection. Carragelose^TM^ acts as a barrier in the nose by forming a gel to trap cold and flu virus particles as they enter the body, thus potentially reducing the amount of virus entering the body and therefore reducing the severity of symptoms [119].

A multicenter, randomized, double-blind, placebo-controlled trial assessing the use of a nasal spray containing iota carrageenan in the prophylaxis of COVID-19 among hospital personnel dedicated to care of COVID-19 patients was carried out in Argentina [120]. The incidence of COVID-19 was significantly lower in the group receiving iota-carrageenan compared to placebo confirming the clinical efficacy of a nasal spray with iota-carrageenan for the prevention of COVID-19.

Antiviral topical nasal spray against COVID-19 under the name Pretz-MD^TM^ saline solution (also known as Nomovid^TM^) was developed by the USA-based Parnell Pharmaceuticals. The company uses the innovative, patented technology derived from the natural herbal extract of the Yerba Santa plant, (*Eriodictyon californicum*). The other essential component of the formulation, MycoDelens is a registered trademark of Parnell Pharmaceuticals. The company claimed that Pretz-MD^TM^ is an easy-to-use and affordable nasal spray product that can be rapidly commercialized over-the-counter to customers. The new formulation is based on a substance licensed by Parnell from New Mexico Tech University, which has the potential to act against drug-resistant bacteria and fungi, such as methicillin-resistant *Staphylococcus aureus* and *Candida auris*. Moreover, the drug is said to act by breaking down the lipids inside the viral envelope and has been tested against the novel coronavirus. The company claimed that its products are natural-based; are patented for oral and nasal care, personal care and anti-infective use; and are commercialized in North America and Europe [121].

### 5.4. Nasal Administration of INNA-051

Australian biotech company ENA Respiratory has proposed a first-in-class nasal spray for the prevention of COVID-19, which has recently entered a Phase I human safety study in Sydney. ENA Respiratory’s therapy, INNA-051, is a small molecule that was developed before the COVID-19 pandemic outbreak and was designed to affect all respiratory infections. INNA-051 is a potent synthetic PEGylated TLR2/6 agonist. It is aimed for intranasal delivery to target the primary entry site of viral respiratory infections including SARS-CoV-2 and influenza, initially infect and replicate in nasal mucosal epithelial cells which express TLR2 and TLR6 on their surface. In preclinical studies INNA-051 demonstrated fast action and significant potential to reduce the time required for nasal epithelial cells to initiate the innate immune responses following virus exposure, providing an advantage to the body in its fight against the virus [122,123]. INNA-051 nasal spray showed minimal systemic bioavailability and pro-inflammatory cytokine release, no direct type I interferon upregulation, durable immune response supporting twice-weekly administration and compatibility with vaccine and intranasal corticosteroids [124,125,126]. The goal of the ongoing randomized, double-blind, placebo-controlled Phase I study is to investigate the safety and tolerability of INNA-051 and to assess pharmacokinetics and pharmacodynamics of the therapy.

### 5.5. Nasal Administration of Ivermectin

Ivermectin is an anti-parasitic agent that has been reported to inhibit SARS-CoV-2 replication in vitro [127] and to cause about 5000-fold reduction in SARS-CoV-2 viral RNA at 48 h. In addition, binding of SARS-CoV-2 spike protein to the human cell membrane may be hindered by Ivermectin docking [128].

In a paper published by Aref et al. [129] the nasal delivery of Ivermectin was discussed. Mucoadhesive nanosuspension was developed as a drug delivery platform to investigate possible effects in mild COVID-19 patients. A prospective clinical trial was carried out involving 114 patients at Qena University Hospital in Egypt, during the period from February to March 2021. COVID-19 patients were treated with Ivermectin mucoadhesive nanosuspension nasal spray. Ivermectin nanosuspension was prepared via nanoprecipitation followed by ultrasonication. Poloxamer 407 and Poloxamer 188 were used as stabilizers. Ivermectin nanosuspension was then incorporated into a mucoadhesive formulation consisting of a mixture of mucoadhesive polymers: HPMC K15M (0.3% *w/v*), Carbopol 974P (0.1% *w/v*) and sodium alginate (0.2% *w/v*). Finally, the formulation was filled into nasal spray containers to allow administration. The study was the first to assess Ivermectin in a nanosuspension formulation for nasal administration. The goal was to reduce the viral load in the upper respiratory tract, providing uniform distribution of the drug through the nasal mucosa. The findings of the study demonstrated significant reduction in hematological and biochemical parameters towards normal values and rapid viral clearance.

Another study, performed by Errecalde et al. [130], used nasal Ivermectin spray to assess its overall safety and local tissue tolerability. The results revealed high tolerability towards the formulation. No macroscopic tissue damage was observed at the application site. Furthermore, histopathological changes were not found in the mucosa or submucosa of the soft palate of the treated animals. Biochemical and hematological parameters were not affected, thus suggesting no risk of adverse effects. Pharmacokinetic assessment of the formulation demonstrated recovery of Ivermectin in plasma, nasopharyngeal and lung tissues following a single-dose application, revealing the highest drug concentrations always measured in nasopharyngeal tissue. Repetitive administration at a 12 h interval resulted into a significant increase in Ivermectin concentrations in nasopharyngeal and lung tissues, both defined targets for SARS-CoV-2, whereas minimal drug systemic exposure was observed. The authors suggest that the intranasal administration of Ivermectin in humans may provide fast, high and persistent concentration at the nasopharyngeal tissue area at doses much lower than those used for oral administration.

### 5.6. Nasal Administration of Chloroquine

The antiviral activity of Chloroquine and Hydroxychloroquine against coronaviruses, and particularly the new SARS-CoV-2, has been demonstrated in cell culture and in animal studies [131,132].

Despite the apparent efficacy in humans with COVID-19, there is some controversy in supporting the data. Efforts have been put into improving response rates and increasing drug dosage; however, the attempts have been limited by enhanced toxicity and mortality. Thakar et al. [133] proposed a nasal treatment of Chloroquine as an alternative to conventional routes of administration. The main goal of the study was to evaluate the safety and efficacy of chloroquine eye drops, commercially available in India, repurposed as nasal drops for prevention of COVID-19 progression. A randomized controlled trial comparing topical administration of chloroquine drops in the nose with standard symptomatic management in patients with mild COVID-19 was carried out at the All India Institute of Medical Sciences (AIIMS), New Delhi in April–May 2020. Six doses of 0.5 mL each were instilled daily for 10 days. The drops were self-administered by the patients at 3-h intervals in the day with a 9-h break at night. Data obtained from the study revealed that Chloroquine was safe and well tolerated after nasal administration. No significant evidence of efficacy was demonstrated in patients with established infection. Favorable virus load trends were noted when administered prior infection, but the findings were limited due to small number of participants in the survey. Further studies with a larger sample size would benefit reaching to a valid conclusion.

### 5.7. Nasal Delivery of Niclosamide

A formulation of Niclosamide, recently identified as a potent inhibitor of SARS-CoV-2 optimized for intranasal administration (UNI91103), was developed the Danish biotech UNION therapeutics. Niclosamide is a broad-spectrum host-targeting antiviral that inhibits viral replication by neutralizing endosomal pH and increases viral clearance stimulating the autophagic flux. UNI91103 (intranasal Niclosamide) is being considered as an alternative to vaccination in high-risk groups, for the prevention of infection. UNI91103 is based on a proprietary salt form of Niclosamide, which is designed for intranasal administration via a nasal spray. According to the results of a recent randomized, placebo-controlled, double-blinded Phase I study, the concentrated Niclosamide solution was well tolerated on nasal administration, meeting the primary endpoints of the trial. Moreover, dose-proportional pharmacokinetic was observed with no indications of systemic accumulation of the drug in the blood [134]. Based on these results, UNI91103 has been selected as the first intervention of the PROTECT-V study (prophylaxis for vulnerable patients at risk of COVID-19 infection), which has recently received UK Urgent Public Heath prioritization. Kidney patients, in particularly those on dialysis, are considered a particularly vulnerable patient population. The trial, led by the Cambridge University Hospitals NHS Foundation Trust and the University of Cambridge, is expected to end by March 2022.

### 5.8. Nasal Delivery of Povidone-Iodine

Povidone-iodine (PI), a complex compound of iodine with the water-soluble polymer polyvinylpyrrolidone, is proven to be effective against wide range of bacteria and viruses, including SARS-CoV [135]. Nasal administration of PI has proven to reduce the viral load in the nasal cavity [136]. In ten investigator-initiated clinical studies, 3M™ skin and nasal antiseptic (PI solution 5% *w/w*, USP) preoperative skin preparation was designed and confirmed effective. A randomized clinical trial investigated the efficacy of nasopharyngeal povidone iodine solutions in reducing the viral load of patients with COVID-19. The main findings of the trial outline the significance of nasopharyngeal decolonization and reduction of the carriage of infectious SARS-CoV-2 in adults with mild-to-moderate COVID-19 [137].

A single-center, open-label randomized clinical trial carried out at Dhaka Medical College Hospital (Dhaka, Bangladesh) was initiated to evaluate the efficacy of povidone iodine against COVID-19 located in the nasopharynx and to assess the adverse events of PI [138]. The study was completed in March 2021, but the results have not been announced yet. An enquiry in Clinicaltrials.gov revealed that multiple trials regarding the potential benefits of PI for COVID-19 are ongoing or have been recently completed [139,140,141,142,143,144,145].

### 5.9. Nasal Delivery of Hypochlorous Acid

Hypochlorous acid (HClO), which is known as a potent broad-spectrum antibacterial agent, has shown strong antimicrobial (antibacterial, fungicidal or virucidal) effect in nasal formulations [146,147].

The Swiss APR Applied Pharma Research S.A., a leader in the development of pharmaceutical drug-delivery technologies and innovative products, have developed a unique Acid-Oxidizing solution (AOS2020) containing pure and stable HClO in a liquid carrier solution, using APR’s Tehclo™ nanotechnology delivery platform, which entraps HCLO in an aqueous solution, enabling its inhalation. AOS2020 comprises a hypotonic solution with unique physicochemical characteristics in terms of pH 2.5–3, oxidative reduction potential 1000–1200 mV and free chlorine species of which pure HClO is not less than 95%. The solution has been already tested for cytotoxicity on fibroblasts, phototoxicity, genotoxicity, vaginal, systemic and ocular irritation. Giarratana et al. [148] evaluated the virucidal efficacy of AOS2020 on human coronavirus SARS-Cov-2 in vitro and the tolerability profile on nasal and oral mucosa. Given the positive results from the study, the authors concluded that AOS2020 has strong potential for nasal and oral treatment of SARS-Cov-2 infection. In May 2021, APR announced the start of a clinical trial to assess the efficacy of its product. The study is being conducted by the Hygiene Unit of IRCCS Policlinico San Martino Hospital (Genoa, Italy). The study involves COVID-19 patients with mild symptoms of the disease and aims at evaluating the efficacy and safety of the spray product in reducing viral load in the upper respiratory airways in recently infected individuals. Results, when available, could represent an additional near-term protective option that could be particularly helpful in high-risk environments.

### 5.10. Covixyl-V

Covixyl-V (Salvacion, USA) is a formulation, based on ethyl lauroyl arginate HCl (LAE), which is a common food preservative known as E243. LAE is a cationic surfactant that has been widely used as an antimicrobial agent in oral-hygiene products. LAE is thought to exert its effects by creating a film on teeth, thus preventing physical attachment of bacteria [149]. It is suggested that a similar mechanism is involved in prevention of COVID-19 infection. Because of the physical barrier created by LAE, viruses are unable to adhere to mucosal tissue of the nasal passages, thus stopping further transmission. This product has been submitted to the FDA for Pre-Emergency Use Authorization and is pending approval for commercial use in the USA [150]. The nasal spray to prevent COVID-19 infection is undergoing human clinical trials to validate the in vitro and in vivo studies performed by SALVACION.

### 5.11. Nasal Delivery of Corticosteroids

According to a study published by Strauss et al. [151], intranasally administered corticosteroids (INCS) could be associated with a lower risk for COVID-19-related hospitalization, admission to the intensive-care unit and hospital mortality. The study was the first of its kind to confirm that INCS had similar impact to that of FDA approved therapies such as remdesivir and systemic steroids. However, the authors conclude that future randomized control trials are needed to determine if INCS reduces the risk for severe outcomes related to COVID-19. Another study [152] was performed to evaluate the severity of rhinological symptoms of COVID-19 by comparison of patients with nasal steroid use and the control group but the conclusions were discouraging suggesting controversial benefit. However, nasal steroids could be effective in the rapid recovery and improvement of olfactory and gustatory dysfunctions of these patients. Such hypothesis has already been placed by Abdelalim et al. [153] who evaluated mometasone furoate nasal spray for its potential in the treatment of post-COVID-19 anosmia. The results of the conducted clinical trial suggested that mometasone furoate nasal spray had no superior benefits regarding duration of anosmia and recovery rates. In another clinical trial safety, the efficacy and tolerability of intranasal dexamethasone as an adjuvant in patients with COVID-19 were investigated. The study demonstrated that dexamethasone administered intranasally reaches the central nervous system through the olfactory nerve and reduces neuroinflammation more effectively than when applied intravenously [154].

Another study presented, for the first time to the scientific community, the preventive capacity of nasal steroids regarding the loss of smell. This effect was attributed to the immunomodulatory role of nasal steroids and their local anti-inflammatory effect around the olfactory nerve [155]. Furthermore, Meneses et al. [156] demonstrated the effectiveness of the intranasal route for the control of peripheral-infection-induced neuroinflammation.

Lipworth et al. [157] hypothesize that INCS, similar to inhaled corticosteroids, cause dose-dependent downregulation of ACE2 expression. It has been hypothesized that people suffering allergic rhinitis, asthma or other respiratory allergies are less susceptible to COVID-19. Indeed, ACE2 expression in airway cells of such people is low, which is associated with decreased sensibility to COVID-19 [158]. However, additional factors beyond ACE2 expression are involved in modulating the response to COVID-19 in allergic individuals. Elucidation of these factors may provide additional important insights into COVID-19 disease pathogenesis. Therefore, further studies are needed to understand the impact of respiratory allergic diseases on COVID-19 severity and susceptibility.

## 6. Conclusions

This review article summarized the current information regarding the intranasal route of administration and its relevance in the combat against the COVID-19 pandemic. Intranasal formulations have been thoroughly investigated for topical and systemic drug delivery, and a vast majority are being explored for their potential against SARS-CoV-2. The nasal route has been proposed as a promising strategy to deliver vaccines and agents known to have antiviral properties against SARS-CoV-2. Tremendous efforts are put into developing safe, efficacious and stable formulations. In the context of the COVID-19 pandemic, it is of crucial importance to have powerful strategies to rely on.

Given the wide variety of therapeutic opportunities offered by the nasal route of administration, and considering the current knowledge, we believe that the time of commercializing of the first reliable nasal formulation against COVID-19 is not very far away.

## Figures and Tables

**Table 1 pharmaceutics-13-01612-t001:** COVID-19 vaccine candidates in clinical and preclinical development [51].

Vaccine Code/Name	Vaccine Type	Developer	Delivery Route	Doses	Development Stage
ChAdOx1-S (AZD1222)	Simian adenovirus vector (spike)	University of Oxford	IN	1 or 2 Day 0 + 28	Clinical trial NCT04816019 Phase I
DelNS1-2019-nCoV-RBD-OPT1	Influenza virus vector	University of Hong Kong Xiamen University Beijing Wantai Biological	IN	2 Day 0 + 28	Clinical trial ChiCTR2000039715 Phase II
COVI-VAC	Live attenuated SARS-CoV-2 virus	Codaginex Serum Institute of India	IN	1 or 2 Day 0 or Day 0 + 28	Clinical trial NCT04619628 Phase I
CIGB-669	Protein subunit	Centre for genetic Engineering and Biotechnology (CIGB)	IN	3 Day 0 + 14 + 28 or Day 0 + 28 + 56	Clinical trial RPCEC00000345 Phase I/II
AdCOVID	Adenovirus vector	Altimmune, Inc.	IN	1 Day 0	Discontinued In Clinical trial NCT04679909 Phase I
Razi Cov Pars	Protein subunit	Razi Vaccine and Serum Research Institute	IM and IN	3 Day 0 + 21 + 51	Clinical trial IRCT20201214049709N2 Phase II
BBV154	Simian adenovirus vector	Bharat Biotech International Limited	IN	1 Day 0	Clinical trial NCT04751682 Phase I
MV-014-212	Live attenuated RSV virus	Meissa Vaccines, Inc.	IN	1 Day 0	Clinical trial NCT04798001 Phase I
Live rNDV vector vaccine	Inactivated virus	Laboratorio Avi-Mex	IM or IN	2 Day 0 + 21	Clinical trial NCT04871737 Phase I
PIV5	Viral vector	CyanVac LLC	IN	1 Day 0	Clinical trial NCT04954287 Phase I
COVAC-ND	Viral vector	Utrecht UniversityWageningen Bioveterinary Research Intravacc	IN	NA	Preclinical
Liposomal formulation with GLA/3M052 adjs.	Protein subunit	University of Virginia	IN	NA	Preclinical
Ad 5 vector	Viral vector	University of Helsinki University of Eastern Finland	IN	NA	Preclinical
Live viral vector based on attenuated Influenza virus	Viral vector	BiOCAD IEM	IN	NA	Preclinical
Recombinant vaccine based on Influenza A virus	Viral vector	FBRI SRC VB VECTOR Rospotrebnadzor Koltsovo	IN	NA	Preclinical
rNDV-LS1-FARVET expressing RBD protein: rNDV-LS1-HN-RBD/SARS-CoV-2	Viral vector	Farmacológicos Veterinarios SAC	IN	NA	Preclinical
rNDV-LS1-FARVET expressing S1 protein: rNDV-LS1-S1-F/SARS-CoV-2	Viral vector	Farmacológicos Veterinarios SAC	IN	NA	Preclinical
mRNA in IN delivery system	mRNA	eTheRNA	IN	NA	Preclinical

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
