# Peer review of "Can the Nasal Cavity Help Tackle COVID-19?"

_pharmaceutics, 2021, doi:10.3390/pharmaceutics13101612_

Round 1
Reviewer 1 Report
In this interesting and detailed review, the authors report the most recent findings regarding the therapeutic agents against COVID-19 pandemic. They focused their attention on nasal administration of vaccines and drugs having antiviral effects against SARS-CoV-2. The manuscript is well written, the authors showed deep knowledge on the topic, reporting a considerable number of references. Therefore, in my opinion, the review could be accepted in Pharmaceutics after minor revision, in order to inform the scientific world on the state of art regarding the vaccines and their administration route.
Minor corrections: pag.3 line 126: studies would be replaced with studied. In addition, in line 127, he would be corrected with the.
Author Response
The authors are grateful to the reviewer for the high evaluation of the manuscript. The remarks have been taken into account and tinput errors have been corrected.
Reviewer 2 Report
【General comments】
In this manuscript, authors review the probability of intranasal vaccine and intranasal therapeutics against COVID-19.
This is a well-described and laudable manuscript.
However there are some points that must be cleared out.
【Specific comments】
(1) Page2, Line59-62
High levels of ACE2 protein in the nasal cells located in the olfactory neuroepithelium were dected. According to “ COVID-19 and bronchial asthma: current perspectives, Masayuki Hojo et.al Global Health & Medicine. 2021; 3(2):67-72” , ACE2 especially, tends to be low in patients with strong atopic factors and in those with poor asthma control. Therefore, it could be speculated that asthma patients are not susceptible to COVID-19.
In the patients of allergic rhinitis, does ACE2 tend to be low, too?
Please let me have a comment .
(2) Page2, Line91-92
Authors write, “The nasal route of administration is considered highly effective for vaccination due to multiple advantages.”
On the other hand, what are disadvantages of the nasal route of administration?
Please let me have a comment .
I hope that my comments are useful for the improvement of this manuscript.
Author Response
Response to Reviewer 2 Comments
We are thankful to Reviewer 2 for the annotation. Our comments are provided below.
Point 1: Page2, Line59-62
High levels of ACE2 protein in the nasal cells located in the olfactory neuroepithelium were dected. According to “COVID-19 and bronchial asthma: current perspectives, Masayuki Hojo et.al Global Health & Medicine. 2021; 3(2):67-72” , ACE2 especially, tends to be low in patients with strong atopic factors and in those with poor asthma control. Therefore, it could be speculated that asthma patients are not susceptible to COVID-19. In the patients of allergic rhinitis, does ACE2 tend to be low, too? Please let me have a comment.
Response 1:
It has been hypothesized that people suffering allergic rhinitis, asthma, or other respiratory allergies are less susceptible to COVID-19. Indeed, ACE2 expression in airway cells of such people is low, which is associated with decreased sensibility to COVID-19 [Reference 158]. A retrospective case–control study performed by Guvey (Guvey A. How does allergic rhinitis impact the severity of COVID-19?: a case-control study [published online ahead of print, 2021 May 1]. Eur Arch Otorhinolaryngol. 2021;1-5. doi:10.1007/s00405-021-06836-z) found that allergic rhinitis did not worsen the severity of COVID-19. There was no significant difference between the case and control groups regarding hospitalization status or length of hospitalization. However, additional factors beyond ACE2 expression are involved in modulating the response to COVID-19 in allergic individuals. Elucidation of these factors may provide additional important insights into COVID-19 disease pathogenesis. Therefore, further studies are needed to understand the impact of respiratory allergic diseases on COVID-19 severity and susceptibility.
This information has been added to the manuscript.
Point 2: Page2, Line91-92
Authors write, “The nasal route of administration is considered highly effective for vaccination due to multiple advantages.” On the other hand, what are disadvantages of the nasal route of administration? Please let me have a comment.
Response 2:
Despite the undeniable advantages of nasal route of administration, there are several limitations that must be considered when designing a nasal dosage form. Apart from the physicochemical properties of the drugs and the dosage forms, a few physiological and pathological conditions associated with the nasal mucosa can also compromise nasal absorption and therapeutic activity. The first step in the nasal route of administration is the administration of the drug to the target area or the optimal site of absorption. Deposition into the vestibule of the nasal cavity may prolong the residence time, but absorption is low; with deposition into the nasal passages, the opposite effect is achieved. The small volume of the nasal cavity limits the amount applied. Subsequently, certain problems may arise when it is necessary to introduce high doses of sparingly water-soluble substances. Another problem is the accuracy and reproducibility of the administered doses, as well as the technique of administration.
Membrane permeability is another limiting factor for the nasal absorption of polar drugs and especially those with high molecular weight. Substances pass through the cell membrane by transcellular transport, using the concentration gradient as a driving force, as well as by receptor-mediated (vesicular) transport or by a paracellular transport mechanism through the tight junctions. In the respiratory part of the nasal cavity, the mucus layer serves as a barrier for membrane permeability, which makes diffusion difficult.
Another important, albeit often overlooked, factor in the low bioavailability, especially of peptides and proteins, is the possibility of enzymatic degradation of molecules in the lumen of the nasal cavity or their passage through the epithelial barrier. The nasal epithelium has a protective enzyme barrier, including the presence of enzymes involved in phases I and II of drug biotransformation. Thus, although GIT and hepatic first-pass metabolism are avoided, there is still a risk of degradation of some substances through the nasal route of administration.
Nasal mucociliary clearance is probably the most important limiting factor for nasal absorption of substances. This natural airway mechanism severely limits the time required for absorption.
The nasal mucosa is highly sensitive to irritation, so toxicity issues are an important limiting factor in the choice of drug substance, excipients and drug delivery system for nasal administration.
To overcome the limitations listed above, special technological approaches are usually applied – permeation enhancers may be used to increase absorption rate, mucoadhesive polymers are employed as carriers to prolong residence time, special devices are developed to assure correct deposition in the nasal cavity, etc.
Reviewer 3 Report
The review manuscript of Pilicheva & Boyuklieva is about the possible future and significance of nasally administered medications against COVID-19. The manuscript is a mixture of a review paper and a concept paper, since most of the discussed preparations are in developmental or clinical phase, the real therapeutic experiences are quite narrow with their applications. Nevertheless, the manuscript is valuable and up to date emphasizing the nasal drug delivery as a potential mode of administration against the COVID-19 pandemic. The work summarizes the currently under development nasal vaccines, antiviral and antiseptic preparations introducing some surprising endeavors like development of hypochlorous acid, povidone-iodine or niclosamide containing nasal preparations.
Few recommendations for minor revision:
- The title must be changed, because it is not covering the content. Nasal cavity cannot tackle with anything, even COVID-19, it is just a cavity. The revised title must involve the nasal administration way and future and -of course- COVID-19, clearly stating that the topic of the review is not a current medication against COVID-19, but the possible therapy of the near future with hope and challenges.
- Row 381. The date mentioned here is expired, perhaps the findings of the trial have been already announced, or the announcement is still delaying. The authors should modify this sentence according to the current situation
- The text should be revised, typos and grammatical errors must be corrected.
Author Response
We highly appreciate the reviewer's valuable comments and recomendations. The responsed are provided below:
- The title must be changed, because it is not covering the content. Nasal cavity cannot tackle with anything, even COVID-19, it is just a cavity. The revised title must involve the nasal administration way and future and -of course- COVID-19, clearly stating that the topic of the review is not a current medication against COVID-19, but the possible therapy of the near future with hope and challenges.
Response: We agree with the reviewer that the nasal cavity cannot tackle anything. However, our goal was to choose a title that sounded provocative, attracted attention and raised questions. That's why we decided not to change the title according to the reviewer's recommendations.
- Row 381. The date mentioned here is expired, perhaps the findings of the trial have been already announced, or the announcement is still delaying. The authors should modify this sentence according to the current situation
Response: The sentence was modified according to the reviewer's comment.
- The text should be revised, typos and grammatical errors must be corrected.
Response: It was done.
Reviewer 4 Report
Manuscript provides a comprehensive and systematic review of COVID-19 key therapeutic strategies employing nasal route of administration. It is well structured and well written and represents the article of great interest to the readership of the journal.
However, there are some issues that need to be addressed by the authors:
Page 1, line 9: “scientists” should be replaced with “for scientists”.
Page 1, line 10: “a great number of” should be replaced with “several”.
Page 3, line 127: “he” should be replaced with “the”.
Page 3, lines 136-138; in this statement (optimal) particle size range might be mentioned.
Page 4, line 163; there are eight vaccines against SARS-CoV2 listed in Table 1 in the category of vaccines in pre-clinical development designed for intranasal administration. Therefore, it seems that in this sentence “seven” should be replaced with “eight”.
Page 6, line 289; “designed” should be replaced with “design”.
Page 12, line 503; authors should revise the following part of the sentence: “(107 virus particles administered intranasally.”.
Page 12, lines 542-544; “The NO produced from NORSTM is totally identical to NO naturally produced by the human body. There is no biochemical, pharmacokinetic, or physical difference between them.” These statements should be revised. Namely, there is no option of two different NO molecules at all.
Page 13, lines 574-576; “None of the patients progressed to severe disease and the number of days to testing positive to negative via COVID-19 RT-PCR nasal swab test was reduced.” It should be clearly stated in relation to which control (referent) group the reduction in number of days to testing positive to negative via COVID-19 RT-PCR nasal swab test was observed.
Page 13, line 595; “ I suggest to replace “used” with “performed”.
Page 15; lines 672 and 678; names “Eriodictyon califonicum”, “Staphylococcus aureus” and “Candida auris” should be written in cursive. The same is suggested for other Latin words like “in vivo” and “in vitro”.
Authors should consider introduction of new section to the manuscript, dealing with the nasal delivery of glucocorticoids in patients with Covid-19, based on information on Randomized Clinical Trial of Intranasal Dexamethasone as an Adjuvant in Patients With COVID-19 (https://clinicaltrials.gov/ct2/show/NCT04513184), and references like Cárdenas et al., Role of Systemic and Nasal Glucocorticoid Treatment in the Regulation of the Inflammatory Response in Patients with SARS-Cov-2 Infection. Arch. Med. Res. 2020, 52, 143–150. and Meneses at al. Intranasal delivery of dexamethasone efficiently controls LPS-induced murine neuroinflammation. Clin. Exp. Immunol. 2017, 190, 304–314.
Author Response
We highly appreciate the Reviewer 4 comments. We have taken into account the remarks:
Page 1, line 9: “scientists” should be replaced with “for scientists”.
Response: It was corrected.
Page 1, line 10: “a great number of” should be replaced with “several”.
Response: It was corrected.
Page 3, line 127: “he” should be replaced with “the”.
Response: It was corrected.
Page 3, lines 136-138; in this statement (optimal) particle size range might be mentioned.
Response: The information was included in the manuscript.
Page 4, line 163; there are eight vaccines against SARS-CoV2 listed in Table 1 in the category of vaccines in pre-clinical development designed for intranasal administration. Therefore, it seems that in this sentence “seven” should be replaced with “eight”.
Response: It was corrected.
Page 6, line 289; “designed” should be replaced with “design”.
Response: It was corrected.
Page 12, line 503; authors should revise the following part of the sentence: “(107 virus particles administered intranasally.”.
Response: It was corrected.
Page 12, lines 542-544; “The NO produced from NORSTM is totally identical to NO naturally produced by the human body. There is no biochemical, pharmacokinetic, or physical difference between them.” These statements should be revised. Namely, there is no option of two different NO molecules at all.
Response: The statement was revised and a part of it was removed.
Page 13, lines 574-576; “None of the patients progressed to severe disease and the number of days to testing positive to negative via COVID-19 RT-PCR nasal swab test was reduced.” It should be clearly stated in relation to which control (referent) group the reduction in number of days to testing positive to negative via COVID-19 RT-PCR nasal swab test was observed.
Response: Unfortunately, such information was not possible to provide clearly since it was lacking in the cited reference. According to data from a systematic review published by Mallet et al. , the duration of respiratory tract virus detection varies greatly within individual participants. In some participants, virus is still detectable at 46 days post-symptom onset. (Mallett, S., Allen, A.J., Graziadio, S. et al. At what times during infection is SARS-CoV-2 detectable and no longer detectable using RT-PCR-based tests? A systematic review of individual participant data. BMC Med 18, 346 (2020). https://doi.org/10.1186/s12916-020-01810-8). The time it takes to fully recover and test negative appears to vary depending on the person, as well as their severity of illness. Some people test negative within 10 days of becoming ill. Others experience a longer period of illness. Some people continue to test positive for weeks, and less commonly; even longer.
Page 13, line 595; “ I suggest to replace “used” with “performed”.
Response: It was corrected.
Page 15; lines 672 and 678; names “Eriodictyon califonicum”, “Staphylococcus aureus” and “Candida auris” should be written in cursive. The same is suggested for other Latin words like “in vivo” and “in vitro”.
Response: It was corrected.
Authors should consider introduction of new section to the manuscript, dealing with the nasal delivery of glucocorticoids in patients with Covid-19, based on information on Randomized Clinical Trial of Intranasal Dexamethasone as an Adjuvant in Patients With COVID-19 (https://clinicaltrials.gov/ct2/show/NCT04513184), and references like Cárdenas et al., Role of Systemic and Nasal Glucocorticoid Treatment in the Regulation of the Inflammatory Response in Patients with SARS-Cov-2 Infection. Arch. Med. Res. 2020, 52, 143–150. and Meneses at al. Intranasal delivery of dexamethasone efficiently controls LPS-induced murine neuroinflammation. Clin. Exp. Immunol. 2017, 190, 304–314.
Response: A new paragraph was added - Line 829, 5.11. Nasal delivery of corticosteroids